# Date, Apple, and Pear By-Products as Functional Ingredients in Pasta: Cooking Quality Attributes and Physicochemical, Rheological, and Sensorial Properties

**DOI:** 10.3390/foods11101393

**Published:** 2022-05-12

**Authors:** Brahim Bchir, Romdhane Karoui, Sabine Danthine, Christophe Blecker, Souhail Besbes, Hamadi Attia

**Affiliations:** 1Laboratory of Analysis Valorization and Food Safety, National Engineering School of Sfax, University of Sfax, Sfax BP W-3038, Tunisia; besbes_souhail@yahoo.fr (S.B.); hamadi.attia@gmail.com (H.A.); 2Higher Institute of Biotechnology of Monastir, University of Monastir, Avenue Taher Hadded BP 74, Monastir 5000, Tunisia; 3Univ. Artois, Univ. Lille, Univ. Littoral Côte D’Opale, Univ. Picardie Jules Verne, Univ. de Liège, INRAE, Junia, UMR-T 1158, BioEcoAgro, F-62300 Lens, France; romdhane.karoui@univ-artois.fr; 4Laboratory of Food Science and Formulation, Gembloux Agro-Bio Tech, University of Liège, Passage des Déportés 2 B, B-5030 Gembloux, Belgium; sabine.danthine@uliege.be (S.D.); christophe.blecker@uliege.be (C.B.)

**Keywords:** pear, date, and apple by-products, pasta, wheat flour, cooking quality, sensorial properties, scanning electron microscopy, multiple factor analysis

## Abstract

This study aims to evaluate the impact of incorporating pear, date, and apple by-products on pasta properties. Pasta properties including cooking quality, texture, color, rheology, thermal gelling, and microstructural characteristics were evaluated. Common wheat flour was substituted by 0, 2.5, 5, 7, and 10 g/100 g of by-products. To choose the best-suited substitute of flour for the preparation of pasta, the sensorial properties of pasta were investigated. Interrelationships between all the physicochemical parameters were investigated using multiple factor analysis. We also studied the impact of storage (7, 15, and 30 days) on the physicochemical proprieties of pasta. The results revealed that the chemical composition of pasta elaborated with by-products was characterized by higher energy (~386 Kcal) and fiber content (~13%) than the control pasta. Generally, materials added to the durum wheat pasta reduce optimum cooking time, adhesiveness, and extensibility, and enhance the swelling index, cooking loss, cooking water absorption, water activity, firmness, and tenacity of pasta. Cooked pasta samples were significantly (*p* < 0.05) darker (L*) and greener (-a*) than the control pasta. Increasing the rate of by-products from 2.5% to 10% principally altered the texture and structure of pasta. Scanning electron microscopy analysis showed that the inclusion of by-products into pasta leads to a disruption of the protein matrix. A practical formulation (2.5% of by-products) can be selected, since a significant difference was detected between overall acceptability scores. Grouping the variables in the principal component analysis plot showed that pasta samples can be divided into three groups. Each group was correlated by a specific variable. A significant modification of the physical parameters of pasta was observed after 30 days of storage.

## 1. Introduction

Health authorities worldwide recommend reducing the consumption of animal proteins and fats and increasing cereal intake, which is a crucial source of dietary fibers [1]. In addition, the World Health Organization considers pasta (a cereal product) as a suitable vehicle for the supplementation of nutrients [2]. Pasta is a traditional food produced using durum wheat flour, and is a staple food in many countries. It is favored by consumers for its ease of transportation, cooking, handling, and storage, as well as its low cost and low glycemic index (2–4) [3]. Nevertheless, traditional pasta is claimed to lack other essential nutritional components such as dietary fibers, vitamins, proteins, minerals, and many other valuable healthy components, such as bioactive molecules [3]. In reality, consumers demand foods with traditional nutritional aspects and additional health benefits.

For the preparation of pasta, it is possible to use non-durum wheat flour and other ingredients (such as dietary fibers) to produce special pasta. By combining the benefits of pasta with the benefits of dietary fibers, new functional food products are created to prevent and to treat diseases such as diabetes and coronary heart disease. Thus, a formulation that supplements pasta with a higher dietary fiber content could enhance fiber intake and decrease the glycemic index of pasta [3]. Different materials have been used as a source of dietary fibers in the preparation of pasta. These materials contain legume flours (bean, chickpea, soybean, pea, and lentil) and flour from other cereals such as amaranth maize, barley, oat, rice, and sorghum [2,3,4,5,6]. The incorporation of dietary fibers into pasta affects the integrity of the protein starch network, hence the tenacity of protein–starch products, and increases pasta quality in terms of optimum cooking time, water absorption, swelling index, texture, taste, appearance, and cooking loss [7].

Some studies suggest adding dietary fruit fiber by-products to wheat flour in pasta formulations due to their nutritional and functional properties, as well as reduced risks of industrial environmental contamination, lower cost, and health benefits [8]. Indeed, different by-products have been used in the formulation of wheat spaghetti, such as tomato and potato pulp by-products [9], carob fibers, carrot pomace [10], unripe banana, and plantain flour [11]. Aguedo et al. [12] showed that the by-products of cooked fruit have specific aromas and high fiber content.

In Belgium, the fabrication of ‘Liege syrup’, a popular fruit concentrate, generates 1000 t of by-products annually from cooked dried dates, pears, and apples [12,13,14]. Agri-food industry by-products are an environmental problem and have repercussions on society and the economy [1]. By-product valorization allows novel ingredients with great nutritional value and healthy properties to be obtained, which can be a potent strategy with which to respond to the demand of consumers for healthier processed foods and to decrease waste [1]. Thus, the development of new products, such as pasta with by-products, can be a strategic area of the food industry. To the best of our knowledge, date, pear, and apple by-products have not been used for the formulation of pasta.

Therefore, the main objectives of this study are to (a) evaluate the impact of incorporating pear, apple, and date by-products on pasta properties, (b) select the best-suited substitute flour for the preparation of pasta, and (c) determine the impact of the storage period test (7, 15, and 30 days) on the physicochemical proprieties of pasta. Multiple factor analysis was used as a statistical tool to analyze the interrelationships between the physicochemical parameters of pasta.

## 2. Materials and Methods

### 2.1. Materials

The basic ingredients used for the formulation of pasta were commercial semolina flour (moisture: 10%; carbohydrate: 78%; protein: 13%; and lipid: 2%), water, and apple, pear, or date by-products. By-products came from dried fruit pomaces, either pears (French and Belgian ‘Conference’ cultivars), apples (Belgian ‘Jonagold’ and ‘Jonagored’, French ‘Granny Smith’ cultivars), or dates (‘Deglet Noor’ cultivar), from the fabrication of ‘Liege syrup’ from the Siroperie Meurens (Belgium). The pomaces were oven-dried for 7 h at 70 °C in a laboratory-scale dryer (Schutzart, Germany, Memmert tcp 800), and ground in a Fritsch laboratory mill with a 1 mm mesh sieve (Haan, Germany). The conservation method and the industrial process of by-products were shown in our previous work [15].

The physicochemical properties of date, apple, and pear by-products are presented in our preceding investigation [15]. The results revealed a predominance of fiber (82% < % of fiber < 91%) in all pomaces, especially insoluble fibers (78% < % of insoluble fibers < 89%), followed by protein (6% < % of protein < 11%), fat (2.5% < % of fat < 3.7%), ash (0.9% < % of ash < 1.4%), and free sugar (0.3% < % of free sugar < 1.1%). Regarding fiber content, it is noteworthy that by-products can be referred to as sources of fiber.

### 2.2. Pasta Manufacturing

The pasta was manufactured according to the method presented by Bouacida et al. [16]. Durum wheat pasta was made using commercial hard wheat semolina (La rose Blanche, Sousse, Tunisia), water, and different types of by-products (apple, pear, or date by-products). By-products were incorporated into recipes by replacing durum wheat flour at the following proportions (*w*/*w*): 2.5%, 5%, 7%, and 10% (*w*/*w*). An additional sample with no by-products included was also made as a control. All dried components of the formula (flour by-products: 70%) were mixed; water (30%) was then added in a Kenwood mixer (Serial, KM 336, Germany) at a short speed for 10 min until the ‘dough’ had an adequate consistency for lamination. After a rest of 1 h, the ‘dough’ was mixed for 10 min at a short speed. The dough was laminated (until it was 2 mm thick) and cut into strips approximately 5 mm wide and 15 cm long using a home-scale-sized pasta lamination machine (LUSSO SP 150, Italy). Then, pasta was dried using a dryer (Memmert tcp 800, Schutzart, Germany) at 45 °C to reach 13 m/m% moisture content, which is recommended by the Codex Alimentarius [17]. The samples were wrapped in cling film and conserved in airtight containers at room temperature for 30 days. The measurements were realized at selected time intervals (at 0, 7, 21, and 30 days).

### 2.3. Chemical Analysis and Nutritional Values

The chemical composition of the pasta was determined by AOAC (1997) methods for ash, lipid, and moisture. The amount of protein was determined using a Dumas Elementar Rapid N cube 161 15054 (Donaustrasse, Germany), as shown in our previous work [14]. The amount of dietary fiber was determined by a theoretical calculation, considering the quantity added to the sample. The amount of carbohydrate was estimated by the difference in mean values, 100-(sum of percentages of ash, moisture, protein, and lipid) [18]. Energy values were evaluated by using the factors 4, 4, and 9 for each gram of protein, ash, and lipid, respectively.

### 2.4. Physical Analysis

#### 2.4.1. Cooking Properties

To determine the cooking properties, pasta samples were analyzed for their optimal cooking time, swelling index, cooking water absorption, and cooking loss. All tests were realized in triplicate.

##### Optimum Cooking Time

The optimum cooking time (OCT) is the time require to reach a complete gelatinization of starch [19]. According to method 16–50 (AACC, 2000), the OCT was determined as the time when the white inner core of the pasta disappeared after cross-cutting it with a razor blade, or after compressing the pasta between two glass slides during 30 s intervals [16].

##### Swelling Index

The swelling index (SI) of cooked pasta (grams of water per gram of dry pasta) was determined by drying pasta samples to a constant weight at 105 °C, expressed as Equation (1):(1)$$\mathrm{SI}=\frac{W_{1}-W_{3}}{W_{3}}$$
where *W*_1_ is the weight of cooked product and *W*_3_ is the weight after drying [20].

##### Cooking Water Absorption

Cooking water absorption (CWA) corresponds to the amount of water that a known dry pasta weight absorbs during cooking and holds after draining (Equation (2)):(2)$$\mathrm{CWA}\;(\%)=\frac{W_{1}-W_{2}}{W_{2}}*100$$
where *W*_1_ is the weight of cooked product and *W*_2_ is the weight of raw pasta [16].

###### Cooking Loss

Cooking loss (CL) is the quantity of dry matter lost in the cooking water under optimal cooking conditions [21]. CL was evaluated by evaporation to a constant weight in an air oven at 105 °C. The residue was weighed and reported as the percentage of the original pasta sample [4].

#### 2.4.2. Quality Measurements

The effect of storage time on pasta qualities was determined during conservation in cling film and stored in airtight containers at room temperature (25 °C). The experiment lasted 4 weeks; samples were taken after 7, 21, and 30 days. During these periods texture, color, and water activity were measured.

##### Texture Measurements

The measurement of the texture of cooked pasta was carried out using a Texture Analyser TAXT2i (Stable Micro System, Watford, UK). The cooked pasta was submitted to compression testing, as shown by Borneo and Aguirre [5]. All pastas were cooked on the day of evaluation. Before testing the pastas, overflow of water was blotted with absorbent paper. Firmness and adhesiveness were calculated by performing a compression test using an AP/36 cylinder probe. The probe compresses the pasta sample by 75% of its original height. The operating conditions of the instrument were as follows: 2 mm/s post-test speed, 2 mm/s test speed, 2 mm/s pre-test speed, and 0.10 N trigger force. Adhesiveness is the force necessary to overcome the attractive force between the surface of the material with which the product comes into contact and the surface of the product, and firmness was determined as the maximum shear strength necessary for the rupture of a sample.

##### Color Measurements

Surface color measurements of pasta were measured according to the method shown by Bchir et al. [22], using a colorimeter (ColorFlex EZ, HunterLab, Reston, VA, USA). The pasta sample color is measured as chromatic ordinates L* (lightness), a* (redness–greenness), and b* (yellowness–blueness) values. Additionally, the total color difference (ΔE) between enriched pasta and control pasta was determined from L*, a*, and b* values.

##### Water Activity

Water activity (a_w_) was measured using an Aqualab Cx-2 instrument (Decagon, Pullman, WA, USA) at 20 °C [22].

#### 2.4.3. Rheological Characteristics

The impact of added by-products on dough rheology characteristics was carried out using an alveograph (Chopin AL 87, France). The 54-30-02 method [23] was employed to measure the alveograph test. The monitored parameters were the dough extensibility (L), the deformation energy (W), the tenacity or resistance to extension (P), and the curve configuration ratio (P/L ratio) of the dough [14].

#### 2.4.4. Differential Scanning Calorimetry

Differential scanning calorimetry (DSC) was performed to determine the thermal gelling properties (gelatinization temperature; temperature onset of gelatinization: T_onset_; gelatinization end point: T_endset_; and total product enthalpy: ∆H) of raw pasta. Thus, to evaluate the influence that by-products might have on the properties of the starch fraction, a TA Instruments Q1000 DSC (New Castle, DE, USA) with a refrigerated cooling accessory and modulated capability was used. Indium and eicosane were used to calibrate the instrument (eicosane, T_onset_: 36.8 °C, ∆H: 247.4 J g^−1^; indium, T_onset_: 156.6 °C, ∆H: 28.7 J g^−1^). Specific heat capacity (Cp) was calibrated using a sapphire. The milled sample (0.5 mm) was first blended with distilled water (1:4) to a total weight of 15 ± 0.3 mg, then sealed hermetically in aluminum pans, and finally left to equilibrate for 1 h prior to the tests. An empty aluminum pan was used as a blank. The temperature range of the scan was 4 and 110 °C with a 10 °C/min heating rate [4].

### 2.5. Sensory Evaluation

The sensory analysis was performed by a hedonic test with 65 untrained participants (32 males and 33 females, aged 20–50 years), as described by Bchir et al. [15]. Each pasta was freshly cooked as per the procedure shown in Section 2.2. After cooking, the pastas were strained, rinsed, and cooled in water at 20 °C. Pastas (100 g) were served on odorless white paper plates with three-digit random number codes. A demographic survey was completed by the participants and evaluations were conducted using a seven-point hedonic scale to determine the degree to which the pasta was appreciated (7 = extremely appreciated, 4 = neither appreciated nor unappreciated, and 1 = extremely unappreciated). Samples were evaluated for the degree of to which they were appreciated for their appearance, color, taste, aftertaste, texture, and overall acceptability.

### 2.6. Scanning Electron Microscopy (SEM)

The microstructure of transversely fractured dough of pasta was performed by scanning electron microscopy (Thermoscientific, Q250, Cambridge, UK). The micrographs were taken using 200× magnification, 70 Pa pressure, and 15.00 KV high voltage.

### 2.7. Statistical Analysis

Statistical analyses were determined using a statistical software program (XLSTAT). Analysis of variance (ANOVA) was performed using Duncan’s test to evaluate significant differences between the samples (*p* < 0.05). To classify the experimental samples of enriched pasta, multiple factor analysis was run using XLSTAT software 2018. Multiple factor analysis transforms the original measured variables into new, uncorrelated variables called principal components.

## 3. Results and Discussion

### 3.1. Chemical Composition of Pasta

The chemical compositions of pasta made with date, apple, and pear by-products are shown in Table 1. Pasta supplemented with date, pear, and apple by-products have a similar composition. Indeed, statistical analysis did not show a significant (*p* > 0.05) difference between pasta enriched with a similar rate of by-products. The chemical composition of pasta was characterized by a high percentage of carbohydrate followed by protein, ash, fiber, and fat. Adding by-product powders decreased the carbohydrate content (from 82 g to ~77%) and increased the fiber (from 4% to ~13%), protein (from 11.55% to ~12.12%), fat (from 1.65% to ~1.91%), and ash (from 4.31% to ~7.95%) fractions compared to the control.

The values of carbohydrate content in enriched pasta were higher than those obtained by Barbara et al. [2] for wheat spaghetti supplemented with silkworm flour (59.5 g/100 g). Protein values of tested pasta were slightly lower than those cited by Chillo et al. [24] and Bouacida et al. [16] for hard wheat spaghetti (13.5 g/ 100 g) and pasta enriched with Eruca leaves (18.74 g/100 g), respectively.

According to Bouacida et al. [16], the value found for fiber (5.50 < % fiber < 13.00%) allows pasta to be classified as a product with a high content of fiber. This could be due to different ingredients, essentially from by-products, used in the formulation of pasta. The amount of fiber in supplemented pasta was in the range of that reported by Borneo and Aguirre [5], Aravind et al. [25], Bouacida et al. [16], and Minarovičová et al. [3], for wheat spaghetti enriched with spinach (4.12%), amaranth (5.79%), spinach leaves flour (4.16%), Eruca vesicaria leaves (10.9%), and pumpkin powder (27.2%), respectively. Table 1 shows that increasing the percentage of by-products in pasta significantly enhanced (*p* < 0.05) the amount of fiber, contrary to fat and protein values, which remained constant.

In addition, Table 1 reveals that fat contents were lower than those found in pasta presented by Borneo and Aguirre [5] (4.86 g/100 g) as well as Bouacida et al. [16] (4.16 g/100 g). This may be explained by the composition of by-products, which are characterized by a low amount of fat (2.5% < fat < 3.7%) [15], which can be an advantage for the food industry to produce low-fat foods. In fact, there is a tendency towards reducing the content of food constituents, such as cholesterol, salt, and fat, which have been related to human health concerns [26]. Table 1 shows that incorporating by-products significantly increased the rate of ash from 4.31% to ~7.95% in pasta (*p* < 0.05). This value is higher than those found by Shreenithee and Prabhasankar [27] in addition to Minarovičová et al. [3] for wheat spaghetti enriched with pea flour (2.56%), wheat flour (0.50%), and pumpkin powder (2.30%).

The incorporation of by-products into pasta reduces the energy of pasta compared to the control (Table 1). Therefore, pasta enriched with by-products could be introduced in the diet food plan. The energy provided by pastas in this work was in the range of that found by Barbara et al. [2] (366–369 Kcal).

### 3.2. Physical Parameters of Pasta

#### 3.2.1. Cooking Properties of Pasta

Cooking properties are a major parameter for the judgment of pasta. Table 2 reveals a significant (*p* < 0.05) decrease in the optimum cooking time (OCT) (from 17.30 to 10.18 min) for all supplemented pasta samples compared to the control. However, the swelling index (SI), cooking water absorption (CWA), and cooking loss (CL) increased significantly (*p* < 0.05), from 2.54 to 3.86 g/g, 98.50 to 123.20, and 3.80 to 6.71 g/100 g, respectively, when the concentration of fiber increased (Table 2). This agrees with the fact that fiber by-products have a higher holding capacity than wheat flour used for the formulation of pasta [14,15]. Results for apple, pear, and date by-product-enriched pasta showed no significant difference (*p* < 0.05) in all cooking parameters when using the same rate of substitution.

Concerning the OCT, Table 2 reveals that the substitution of wheat flour with by-products significantly reduced OCT from 17.30 min (control) to ~10.18 min (10% by-products). Furthermore, the increase in the by-product rate from 2.5% to 10% considerably reduced the OCT from 16.20 min to 10.18 min, respectively. These results are in concordance with those obtained by Kuchtová et al. [28], Petitot et al. [29], and Bouacida et al. [16] after the addition of wheat flour to pasta with Vicia faba, pumpkin, bean flour, eruca vesicaria leaves, and oat bran powder, respectively. Gluten is a major ingredient responsible for the development of the starch–protein structure. Therefore, a dilution of these components with fiber by-products reduces the OCT, as shown in [1,30]. According to Lucas-Gonzalez et al. [1], reducing the OCT of spaghetti could provide the market with products that require less processing time.

Table 2 shows that the supplementation of by-products can lead to higher swelling and CWA of the pasta (Table 2). A great rate of substitution with by-products is associated with a very high swelling capacity and water absorption, reflecting the effect of the fiber on the swelling power of the samples.

Regarding the SI, Table 2 shows that its values ranged from 2.54% (for the control) to 3.86% (for the pasta containing 10% by-products). The greatest SI was obtained for pasta containing 10% by-products. This may be due to the great capacity of fibers to absorb and retain water within a very-well-developed starch–protein–polysaccharide network. The results support those found by Tudorica et al. [4] and Bouacida et al. [16], showing the effect of pea fiber and Eruca vesicaria leaves powder on the CL and SI of pasta.

CWA indicates the amount of water absorbed by the pasta during cooking [31]. According to the MSZ 20500/1–1985 standard, CWA must be at least 100% of the dry pasta mass. All formulations of pasta achieved the minimum requirement. It was shown that the addition of by-products from 2.5% to 10% gradually increased CWA from 98.0% (control) to 123.2% (10%). The pasta with the highest quantity of by-products showed the highest CWA. These observations can be explained by the competition between the starch and fiber for the absorption of water [30]. These results are consistent with those obtained by Rosa-Sibakov et al. [32] (faba starch), Kuchtová et al. [28] (pumpkin powder), and Aravind et al. [25] (guar gum and carboxymethylcellulose).

CL is one of the most important parameters that can predict the overall pasta cooking performance by both industry and consumers, with a low value showing good quality [33]. Results revealed that the CL for all fiber-enriched samples was slightly higher than that of the control (3.80%). According to Table 2, the CL increased from 4.04% to 6.71% as the rate of by-products was increased from 2.5% to 10%. These values are close to those obtained by Wang et al. [34], showing an enhancement of CL from 4.95% to 7.57% when the rate of rice bran fiber was ~15%. The CL values are similar to those observed for pasta supplemented with pea fibers (5.77% < CL < 6.99%) [4]. By-product-enriched pastas are judged to be of excellent quality because the CL was smaller than 12% for all rates of addition [35]. The results agree with those obtained by Kaur et al. [36], who exposed a positive linear correlation between the CL and the rate of bran addition. The increase in CL may be due to a disruption in the protein–starch matrix and the uneven distribution of water within the pasta matrix due to the competitive hydration tendency of fiber by-products. This explanation agrees with the results reported by Tudorica et al. [4], who exposed the interactions of pea fibers in pasta products. Therefore, the presence of by-products disrupts the protein network and increases the CL.

#### 3.2.2. Texture Analysis

The mean values of adhesiveness and firmness for cooked pasta are summarized in Table 3. Statistical analysis showed that all textural parameters were affected by the addition of by-products (*p* < 0.05) as well as during storage. In addition, as shown in Table 3, pastas with enriched apple, pear, and date by-products have similar textural properties. The increase in by-product supplementation caused a decrease in firmness (from 12.50 N to 7.24 N) and an increase in the adhesiveness (from −0.96 N.s to −0.17 N.s) compared with the formulation with no fiber by-products. The results are in line with those exposed by Silva et al. [37], showing that the additional rates of 5 and 10% with barley and oat bran formulations seemed to cause a decrease in pasta firmness. Texture analysis shows that the firmness of pasta supplemented with by-products was lower than that of the control (Table 3). This would suggest that by-products destabilize the structure strength of pasta. This hypothesis could also be related to the high values obtained for CL, showing a destroyed structure from which large rates of solids are produced during cooking. This event was exposed by Wojtowicz and Moscicki [38] as well as Bouacida et al. [16]. Bustos et al. [6] showed that the extra oat bran disrupted the formation of the protein–starch matrix of the pasta, leading to lower values of firmness. The decrease in pasta firmness could be associated with the role of non-gluten proteins or the insoluble fiber present in the plant material (by-products), which might interfere with the continuity of the gluten matrix, probably making it weaker [25,39].

The pastas enriched with by-products were less sticky than the control. A similar trend was appreciated in vermicelli adhesiveness values after the addition of 20 g/100 g of wheat bran [40] as well as in fresh pasta supplemented with inulin and pea fiber [41]. Statistical analysis showed that while firmness decreased adhesiveness increased significantly (*p* < 0.05) during conservation (Table 2). Similar observations are reported by Borneo and Aguirre [5] as well as Bouacida et al. [16]. This may be due to the high amount of fiber in by-products, which influenced the texture of pasta.

#### 3.2.3. Color Characteristics of Pasta

CIELAB coordinates (L*a*b*) of all pastas are indicated in Table 4. The L*, a*, and b* values of the control (0% by-product) were 70.20 ± 2.23, 5.93 ± 0.15, and 25.54 ± 1.25, respectively. These results showed that the control sample has higher L*, a*, and b* values. Islas-Rubio et al. [39] showed that pasta products made from semolina with high L* and b* values induce a more desirable product. Similar observations were obtained by Bouacida et al. [16]. Indeed, pasta enriched with by-products induced a decrease in all color parameters from 70.20 ± 2.23, 5.93 ± 0.15, and 25.54 ± 1.25 to ~41, ~−1, and ~8 for L*, a*, and b*, respectively. This is in concordance with the findings of Abdel-Moemin et al. [7]. By-products have an apparent effect on the a* value, leading to a greener hue in the pasta. This enhancement in greenness could be due to the lignin component that composes fiber; lignin has an aromatic structure and might cause more Maillard reactions with other components in the food matrix [6]. Therefore, the pastas enriched with by-products have a darker tint. The results obtained in the present study are in concordance with data mentioned in the literature; specifically, pasta enriched with dietary fiber was significantly darker (lower L*) [42].

In addition, results showed that an increase in the rate of by-products causes cooked pasta to be darker (decrease in L* value) (Table 4). CIELAB coordinates ‘b*’ (positive values) and ‘a*’ (negative values) (Table 2) significantly decreased due to pasta ingredients, essentially the color of the by-products, and were mainly associated with caramelization and Maillard reactions during the preparation of cooked pasta. In addition, browning reactions could also be due to the deterioration of unsaturated galacturonides in pectin that happened during the drying step [42].

Results did not reveal a significant difference between the samples according to the storage time for all color parameters. This reveals that the color of pasta was stable during all periods of conservation. Comparable observations were reported by Cemin et al. [43] for pasta supplemented with spinach and broccoli.

Furthermore, the color difference (ΔE) increased significantly (*p* < 0.05) as the rate of by-products was increased from 2.5% to 10%, which agrees with Barbara et al. [2]. According to the color difference, the results of pasta revealed an observable difference (ΔE > 5) amongst all samples. This shows the presence of two different colors [44]. The color changes of the dry pasta during storage could be due to enzymatic processes and non-enzymatic oxidation [2].

Generally, it could be concluded that all fiber source materials altered the L*, a*, and b* values, resulting in all formulations having a more brownish hue compared to the reference (control).

#### 3.2.4. Water Activity Characteristics of Dried Pasta

Table 1 shows that the a_w_ measurements of all by-product-supplemented pasta samples were overall higher than that of the control. This tendency could be due to higher water absorption by fibers. Indeed, several authors argue that the dietary fibers supplemented into food products can enhance oil and water holding capacity, emulsification, and/or gel formation [45,46]. Indeed, dietary fiber structures present an important number of hydroxyl groups which provide more water interactions through hydrogen bonding [47]. Table 5 shows that enriched pastas have similar a_w_ values. This could be due to the similar chemical composition of all fibers from cooked fruit by-products [15]. In addition, Table 5 shows that a_w_ increased as the rate of fibers increased from 2.5% to 10%. Results obtained in this study are parallel to those reported by Sudha et al. [40] at 5, 10, and 15 percent substitution rates of oat bran in pasta. After 30 days of storage all formulated pastas had noticeably lower a_w_ rates, ranging from 0.537 to 0.570, compared to the beginning of the process (0.701 < aw < 0.672). This fact is due to water loss during the conservation process. Moreover, statistical analyses revealed a significant difference (*p* < 0.05) between all formulated pastas during storage. The water activity (a_w_) values observed in the current work are in accordance with those reported by Aramouni and Mahmoud [48], Sun-Waterhouse and Wadhwa, [46] and Silva et al. [37], showing that the a_w_ of pasta varied between 0.226–0.650.

Table 5 shows that the water activity of pasta enriched with dietary fibers remained higher than that of the control after the storage time. The a_w_ values show that all the dried pastas produced in this study would have a good shelf life. In fact, all a_w_ rates are below 0.650; thus, enriched pasta should be stable against microbial growth and might persist for about six months [48].

#### 3.2.5. Rheological Properties of Enriched Pasta

The effect of apple, pear, and date by-product addition at different rates on the alveograph parameters of pasta dough are shown in Table 6. The addition of by-products always significantly (*p* < 0.05) enhances dough tenacity (P) (from 80.23 to ~135 mm of H_2_O) and significantly reduces dough extensibility (L) (from 62.53 to ~17 mm) compared to the control. Therefore, doughs have the ability to retain gas with a low capacity of extending without breaking down. The enhancement of P values is likely due to the interaction between flour protein and fiber structure, or the bad hydration of doughs supplemented with by-products [34]. In fact, by-product-enriched pastas require more water than the control due to their significant number of hydroxyl groups which allow more water interactions. The determination of the value of the P/L ratio gives information about the extensibility and elastic resistance balance of flour dough and summarizes the effect of L and P parameters. Table 6 shows that the value of the P/L ratio increased (from 1.29 to 7.72) as the rate of fibers increased from 2.5% to 10%. Wang et al. [34] claim that an increase in the P/L parameter might be caused by the high content of cellulose present in the fibers, which promotes a powerful interaction between fibers and flour protein. Table 6 shows a statistical difference (*p* < 0.05) between the enriched pasta and the control concerning the deformation energy. In fact, the dough deformation energy decreased significantly (*p* < 0.05) (from 154 × 10^−4^ to ~124 × 10^−4^) with the addition of by-products. Similar observations are reported by Shreenithee and Prabhasankar, [27] revealing that the addition of yellow pea flour reduces the dough deformation energy.

#### 3.2.6. Thermal Gelling Properties of Pasta

The impacts of the inclusion of apple, date, and pear by-products on starch gelatinization properties were investigated using DSC methodology. Results from DSC showed that the addition of by-products affects the thermal gelling properties of pasta (Table 3). The starch gelatinization temperature of enriched pasta decreased proportionally as the rate of by-products was raised from 2.5% to 10%. As such, the results are consistent with previous research [4], which reveals that the addition of soluble non-starch polysaccharides reduces gelatinization temperature. This is partly due to the soluble fibers competing with starch for water absorption and therefore limiting gelatinization as well as starch swelling events, resulting in a lower than expected T_endset_ value (Table 7). Martín-Esparza et al. [49] revealed a similar result. If fact, the authors found that incorporating tiger nut flour into pasta reduced the starch gelatinization temperature due to the presence of a smaller amount of starch available for gelatinization; therefore, a lower amount of energy is needed for such a transition to occur.

Table 7 shows that the enthalpy of an enriched sample decreased with an increase in the rate of fibers. This rise was not moderate and showed significant (*p* < 0.05) differences between the control and fibers from different origins. The enthalpy of a system is an indicator of the quantity of starch gelatinization within a flour or starch base, and should therefore be linked to the gelatinization temperature of starch. One possible explanation for the pattern of enthalpy values of fiber systems is that, at a high rate of by-product, the fiber greatly competes for water with starch, affecting gelling and pasting events. Therefore, the addition of apple, pear, and date by-products to pasta has a significant (*p* < 0.05) impact on starch gelatinization properties.

### 3.3. Scanning Electron Microscopy of Pasta Dough

The internal structure of pasta dough is shown in Figure 1. The micrograph of the control dough sample shows that the protein–starch matrix is well-formed, with continuous and strong protein strands entrapping large starch granules. In fact, the control sample presents swollen and gelatinized starch granules which appear to be integrated into a developed protein matrix to make a compact structure, with some starch granules not gelatinized. These results are in accordance with those of Bustos et al. [6].

Micrographs of pasta dough with the inclusion of by-products show a similar protein–fiber matrix for all types of by-product-enriched pastas. The supplementation of by-products into the pasta involves a disruption of the continuity of the protein matrix. The protein–fiber matrix within pastas containing by-products at 5%, 7.5%, and 10% appears to be less developed than that of the control, resulting in an open appearance with discrete starch granules ‘uncovered’ and exposed to enzymatic attack [6]. The degree of disruption appears to increase with the rate of fibers added to the product. However, the addition of by-products at 2.5% appears not to significantly affect pasta structure compared with that of the control. These results are in accordance with those obtained by replaced Ainsa et al. [19] and Bustos et al. [6]. This disruption to the pasta structure may explain the decrease in firmness observed in Table 3. A modified pasta microstructure can affect the rate of starch degradation which affects the insulinemic and glycemic indices [6].

### 3.4. Sensory Evaluation

The sensory properties of cooked pasta enriched with apple, pear, and date by-products were evaluated to choose the best-suited substitute flour for the formulation of pasta. The mean scores of sensory attributes are shown in Table 8. Statistical analysis did not show a significant difference (*p* > 0.05) for taste scores between all pasta formulations. Therefore, consumers could not differentiate the taste of pasta between different percentages of substitution.

For other sensory attributes, there is a significant difference between different amounts from 5% of added by-product. Hence, the consumer cannot detect the difference in the aftertaste, appearance, texture, and color of pasta between the control pasta and that enriched with 2.5% of added by-product. Therefore, consumers reacted in different ways to all formulations. These results agree with previous SEM observations, showing that the addition of 2.5% of by-product did not significantly affect the structure of the pasta dough compared to the control.

The pasta enriched with 2.5% of by-product presented the highest overall acceptability score. Table 8 indicates that the overall acceptability decreased significantly (*p* < 0.05) as the amount of by-product was increased from 5% to 10%. Therefore, consumers did not appreciate pasta containing a high by-product rate (above 5%). Statistical analysis revealed a significant (*p* < 0.05) difference in overall acceptability between the pasta supplemented with 2.5% of by-product and the other samples (Table 8). In the same way, Sant’Anna et al. [50] and Crizel et al. [8] revealed that fettuccini pasta with the addition of a higher amount of (75 g/kg, 50 g/kg, and 25 g/kg) grape marc and orange by-product fiber powder resulted in lower acceptability.

The textural and color characteristics of food products have a major role in the final acceptance by consumers. Table 8 shows a significant decrease in texture and color scores with an increase in fiber content in pasta. Indeed, our previous investigations showed that apple, pear, and date by-products have low color parameter (L*, a*, and b*) values, inducing a dark color [15]. Therefore, the original colors of by-products significantly affect the appreciation of the consumer.

Therefore, a practical formulation can be chosen, since a significant difference (*p* < 0.05) was shown between overall acceptability scores. In fact, panels have appreciated the pasta enriched with only 2.5% of by-product more than the other formulations.

### 3.5. Multiple Factor Analysis (MFA)

Physicochemical and sensorial results were subjected to multiple factor analysis (MFA). The MFA plot of pasta is illustrated in Figure 2A,B, which describes the (A) interrelations among analyzed parameters and the (B) positioning of analyzed pastas in comparison to each other, respectively. The first axis (F1) accounted for 59.16% of the total variance and the second (F2) for 19.83%, accounting for 79.00% of the total variance. Therefore, the first two axes have almost 70% of the variability. From this plot, we can expose that F1 axis was positively correlated (localized in the positive axis of F1) with sensory attributes, optimum cooking time, deformation energy, extensibility, and firmness. On the other hand, the variables taste, swelling index, deformation energy, cooking loss, gelatinization temperature, and textural parameters (tenacity, extensibility, and firmness), were positively loaded with the second axes (F2). Therefore, textural parameters (extensibility and firmness), taste, and gelatinization temperature were positively correlated with both axes.

According to the results of the MFA, the pasta samples were divided into three well-defined groups: (1) pasta supplemented with 2.5% of apple, pear, and date fiber by-products, (2) control (0% fiber by-products) (3) all the other formulations (Figure 2B). This corroborates our previous observations showing that the addition of by-products induces a modification of the physicochemical properties of the pasta. Indeed, the control sample is the furthest to the bottom right, and the other samples are located at the upper left position. This confirms our observation, showing that supplemented pastas have a different sensory profile compared to the control.

The combination of the results of Figure 2A,B show that sensory attributes characterize the first group (pasta supplemented with 2.5% of by-products). However, textural parameters (extensibility and firmness) characterize the second group (the control).

According to the results of MFA, extensibility and firmness represent two closest variables (RV: 0.9911). In fact, RV coefficients show the relationship between variables (1: high correlation; 0: low correlation). In addition, cooking water absorption–tenacity (RV: 0.9368), adhesiveness–extensibility (RV: 0.9703), and tenacity–extensibility (RV: 0.9385) constitute closest variables.

In addition, the sensory attributes and principally the overall acceptability were grouped on the right side of the plot, and the pasta supplemented with 5%, 7.5%, and 10% of by-products were grouped on the left side of the MFA plot. The visualization of these factors reveals that a higher by-product rate negatively affects the appreciation of the consumer. This validates our previous results which showed that consumers attribute the highest score to pasta fortified with 2.5% of by-products.

## 4. Conclusions

From the overall results, it could be concluded that pasta can be formulated with flour from pear, apple, and date by-products. Indeed, by-products have a positive impact on physicochemical properties and cooking quality attributes of pastas. The addition of by-products improves the fiber content, swelling index, and cooking water absorption of pastas, while reducing their optimum cooking time. However, enhancing the by-product rate (from 2.5% to 10%) negatively alters the texture and the structure of pasta. Indeed, SEM analysis showed that the inclusion of by-products into pasta disrupts the protein matrix. Moreover, the pasta fortified with by-products exhibited a darker color than that of the control semolina pasta. Pear, apple, and date by-products could be added to pasta at a rate of 2.5%. They are considered as the most acceptable organoleptically, with the highest overall appreciation score. This means, that by-products are a suitable material for enriching pasta. In addition, MFA results revealed that the pastas enriched with 2.5% of by-products are distinguished, from the other samples, by their sensory attributes. We also carried out a storage test, which showed a significant modification of the texture and water activity after 30 days, contrary to the color of pasta, which remained stable. Additionally, further research can be carried out on the different shapes of pasta and their effects on the quality of the product as well as the influence of additives in improving the texture of pastas.

## Figures and Tables

**Figure 1 foods-11-01393-f001:**
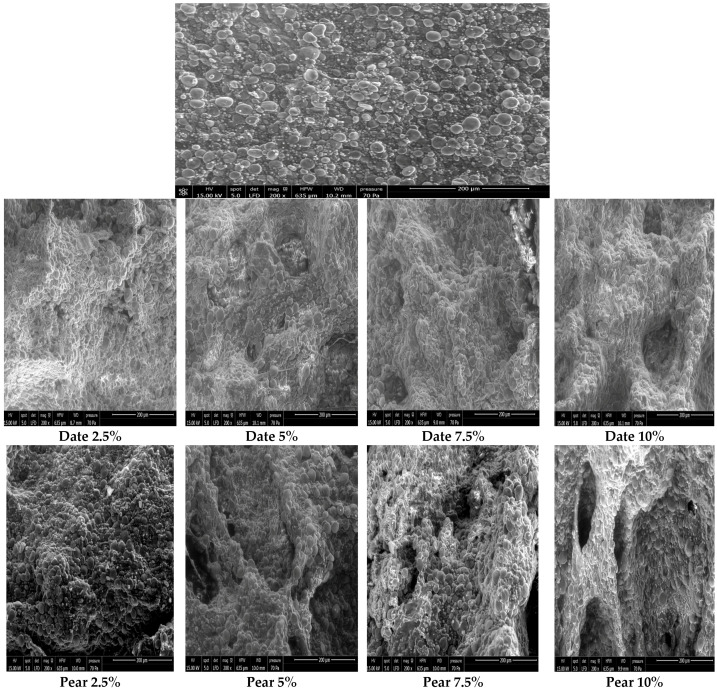
Scanning electron microscopy of dough enriched with date, apple, and pear by-products.

**Figure 2 foods-11-01393-f002:**
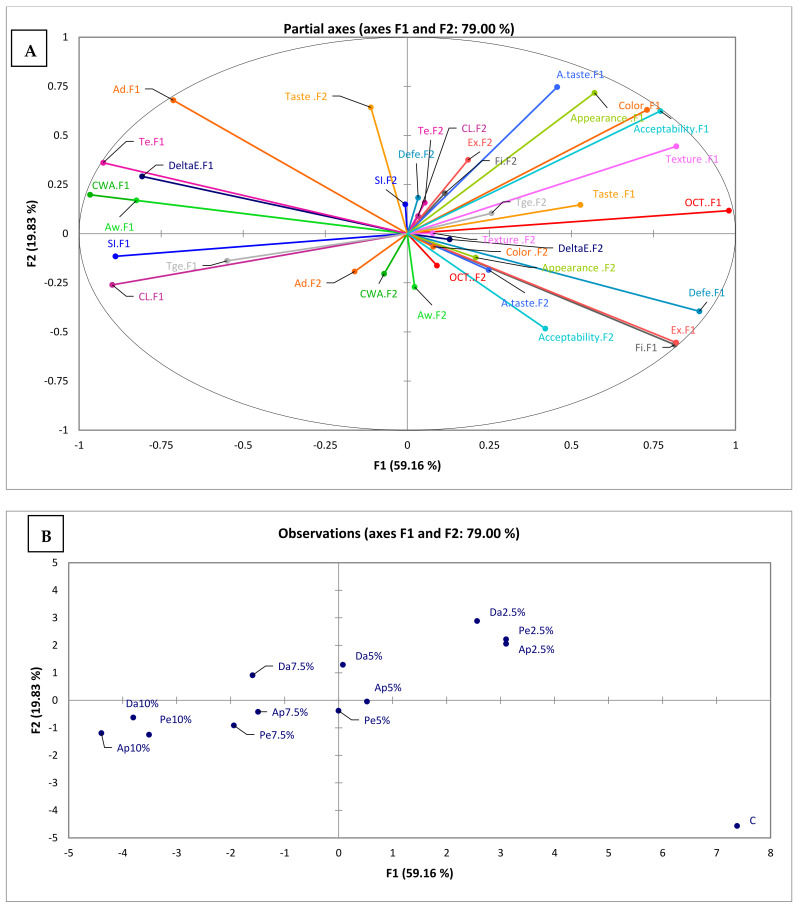
(**A**) The score and loading plots for F1 versus F2 (OCT: optimum cooking time; SI: swelling index; CWA: cooking water absorption; CL: cooking loss; Fi: firmness; Ad: adhesiveness; DeltaE: color difference (ΔE); Aw: water activity; Te: tenacity; Ex: extensibility; Defe: deformation energy; and Tge: gelatinization temperature). (**B**) The distribution of different samples in relation to the F1 and F2 axes.

**Table 1 foods-11-01393-t001:** Chemical composition of pasta and calculated nutritional values.

By-Product Addition (g/100 g Pasta)		Fiber (g/100 g)	Protein (g/100 g)	Carbohydrate (g/100 g)	Fat (g/100 g)	Ash (g/100 g)	Energy (Kcal)
**Apple**	2.5%	5.30 ± 0.01 ^e^	11.79 ± 0.31 ^a^	80.23 ± 1.10 ^abc^	1.70 ± 0.21 ^a^	6.28 ± 0.15 ^de^	383.38 ± 3.22 ^bc^
	5%	7.85 ± 0.25 ^d^	11.99 ± 0.22 ^a^	79.78 ± 1.15 ^abc^	1.82 ± 0.21 ^a^	6.41 ± 0.10 ^d^	383.46 ± 1.15 ^bcd^
	7.5%	10.08 ± 0.10 ^c^	12.03 ± 0.51 ^a^	78.59 ± 2.02 ^bc^	1.86 ± 0.15 ^a^	7.52 ± 0.50 ^abc^	379.22 ± 2.54 ^cde^
	10%	12.42 ± 0.20 ^b^	12.12 ± 0.61 ^a^	77.81 ± 2.50 ^c^	1.88 ± 0.10 ^a^	8.19 ± 0.20 ^a^	376.64 ± 2.31 ^e^
**Pear**	2.5%	5.60 ± 0.35 ^e^	11.66 ± 0.55 ^a^	80.36 ± 0.23 ^abc^	1.71 ± 0.14 ^a^	6.27 ± 0.22 ^de^	383.47 ± 2.10 ^bc^
	5%	7.29 ± 0.50 ^d^	11.83 ± 0.10 ^a^	79.18 ± 1.50 ^bc^	1.80 ± 0.15 ^a^	7.19 ± 0.25 ^c^	380.24 ± 1.86 ^cde^
	7.5%	10.28 ± 0.01 ^c^	12.08 ± 1.02 ^a^	78.59 ± 1.20 ^bc^	1.91 ± 0.30 ^a^	7.42 ± 0.70 ^bc^	379.87 ± 3.18 ^cde^
	10%	13.00 ± 0.61 ^a^	12.01 ± 0.15 ^a^	78.20 ± 1.15 ^bc^	1.93 ± 0.01 ^a^	7.77 ± 0.10 ^abc^	378.57 ± 3.71 ^de^
**Date**	2.5%	5.80 ± 0.55 ^e^	11.79 ± 0.10 ^a^	80.86 ± 2.15 ^ab^	1.65 ± 0.21 ^a^	5.70 ± 0.51 ^e^	385.45 ± 2.50 ^d^
	5%	7.45 ± 0.41 ^d^	11.98 ± 0.21 ^a^	79.80 ± 0.25 ^abc^	1.70 ± 0.11 ^a^	6.52 ± 0.45 ^d^	382.42 ± 1.15 ^bcd^
	7.5%	10.20 ± 0.10 ^c^	12.05 ± 0.31 ^a^	78.86 ± 2.50 ^bc^	1.75 ± 0.15 ^a^	7.34 ± 0.42 ^abc^	379.39 ± 2.40 ^cde^
	10%	12.70 ± 0.30 ^ab^	12.11 ± 0.45 ^a^	78.12 ± 1.15 ^bc^	1.82 ± 0.40 ^a^	7.95 ± 0.10 ^ab^	377.30 ± 0.56 ^de^
**Control**	0%	4.00 ±0.01 ^f^	11.55 ± 0.60 ^a^	82.49 ± 1.50 ^a^	1.65 ± 0.20 ^a^	4.31 ± 0.25 ^f^	391.01 ± 1.23 ^a^
** *F* **		**266.59**	**0.46**	**2.23**	**0.71**	**30.27**	**7.80**

Means in the same column with different letters are significantly different (*p* < 0.05).

**Table 2 foods-11-01393-t002:** Cooking properties of enriched pasta.

By-Product Addition (g/100 g DM)		Optimum Cooking Time (Min)	Swelling Index (g of Water/g of Pasta)	Cooking Water Absorption (g/kg)	Cooking Loss (g/100 g of Pasta)
**Apple**	2.5%	16.20 ± 0.01 ^b^	2.94 ± 0.12 ^de^	111.00 ± 2.50 ^h^	4.06 ± 0.15 ^cd^
	5%	13.32 ± 0.04 ^f^	3.05 ± 0.01 ^cd^	114.10 ± 2.15 ^fg^	4.55 ± 0.52 ^c^
	7.5%	12.47 ± 0.04 ^g^	3.25 ± 0.05 ^bc^	120.32 ± 0.05 ^bcd^	5.10 ± 0.11 ^b^
	10%	10.51 ± 0.01 ^j^	3.86 ± 0.04 ^a^	123.20 ± 1.15 ^a^	6.41 ± 0.16 ^a^
**Pear**	2.5%	15.06 ± 0.02 ^d^	2.82 ± 0.13 ^e^	113.50 ± 1.50 ^gh^	4.10 ± 0.10 ^cd^
	5%	13.54 ± 0.03 ^e^	2.95 ± 0.02 ^de^	116.41 ± 2.12 ^ef^	4.21 ± 0.13 ^c^
	7.5%	12.26 ± 0.01 ^h^	3.14 ± 0.11 ^bcd^	119.64 ± 1.15 ^cd^	5.23 ± 0.40 ^b^
	10%	11.15 ± 0.24 ^i^	3.36 ± 0.22 ^b^	121.23 ± 1.50 ^abc^	6.20 ± 0.24 ^a^
**Date**	2.5%	15.19 ± 0.02 ^c^	2.77 ± 0.15 ^e^	112.00 ± 1.18 ^gh^	4.04 ± 0.41 ^cd^
	5%	13.46 ± 0.03 ^e^	2.90 ± 0.14 ^de^	115.10 ± 1.50 ^ef^	4.40 ± 0.55 ^c^
	7.5%	12.27 ± 0.05 ^h^	3.10 ± 0.22 ^cd^	118.23 ± 1.30 ^de^	5.15 ± 0.32 ^b^
	10%	10.18 ± 0.02 ^k^	3.23 ± 0.13 ^bc^	122.10 ± 1.52 ^ab^	6.71 ± 0.17 ^a^
**Control**	0%	17.30 ± 0.01 ^a^	2.54 ± 0.01 ^f^	98.50 ± 1.50 ^i^	3.80 ± 0.22 ^d^
** *F* **		**3181.54**	**21.63**	**58.27**	**33.46**

Means in the same column with different letters are significantly different (*p* < 0.05): DM: dry matter.

**Table 3 foods-11-01393-t003:** Firmness and adhesiveness of cooked pasta made with added apple, pear, and date by-products.

By-Product Addition (g/100 g DM)			T_0_	T_1_	T_2_	T_3_	*F*
**Apple**	** *Firmness* ** ** *(N)* **	**2.5%**	8.25 ± 0.20 ^a^	8.00 ± 0.11 ^ab^	7.55 ± 0.05 ^b^	6.93 ± 0.05 ^c^	**29.88**
		**5%**	8.05 ± 0.10 ^a^	7.25 ± 0.12 ^ab^	6.86 ± 0.05 ^b^	6.04 ± 0.01 ^b^	**4.263**
		**7.5%**	7.65 ± 0.21 ^a^	6.35 ± 0.05 ^b^	6.02 ± 0.21 ^c^	5.85 ± 0.15 ^c^	**95.88**
		**10%**	7.25 ± 0.13 ^a^	6.21 ± 0.10 ^b^	5.86 ± 0.15 ^c^	5.10 ± 0.12 ^d^	**154.6**
	** *Adhesiveness (N.s)* **	**2.5%**	−0.24 ± 0.01 ^a^	−0.22 ± 0.02 ^a^	−0.20 ± 0.01 ^a^	−0.19 ± 0.01 ^a^	**0.55**
		**5%**	−0.22 ± 0.00 ^c^	−0.19 ± 0.01 ^b^	−0.16 ± 0.02 ^a^	−0.14 ± 0.01 ^a^	**24.50**
		**7.5%**	−0.20 ± 0.02 ^c^	−0.15 ± 0.01 ^b^	−0.13 ± 0.01 ^ab^	−0.11 ± 0.02 ^a^	**17.90**
		**10%**	−0.18 ± 0.01 ^c^	−0.14 ± 0.01 ^b^	−0.12 ± 0.02 ^ab^	−0.10 ± 0.02 ^a^	**14.00**
**Pear**	** *Firmness* ** ** *(N)* **	**2.5%**	8.46 ± 0.15 ^a^	7.22 ± 0.02 ^b^	6.85 ± 0.10 ^c^	6.51 ± 0.18 ^d^	**132.13**
		**5%**	8.26 ± 0.20 ^a^	7.13 ± 0.15 ^b^	6.64 ± 0.01 ^c^	6.05 ± 0.15 ^c^	**30.86**
		**7.5%**	8.05 ± 0.10 ^a^	7.02 ± 0.10 ^b^	6.45 ± 0.16 ^c^	5.85 ± 0.10 ^d^	**192.28**
		**10%**	7.55 ± 0.50 ^a^	6.95 ± 0.30 ^b^	6.01 ± 0.12 ^c^	5.65 ± 0.10 ^c^	**63.53**
	** *Adhesiveness* ** ** *(N.s)* **	**2.5%**	−0.25 ± 0.01 ^b^	−0.21 ± 0.05 ^ab^	−0.20 ± 0.01 ^ab^	−0.17 ± 0.01 ^a^	**5.17**
		**5%**	−0.23 ± 0.01 ^c^	−0.19 ± 0.03 ^b^	−0.17 ± 0.01 ^ab^	−0.15 ± 0.02 ^a^	**9.33**
		**7.5%**	−0.21 ± 0.01 ^b^	−0.20 ± 0.01 ^b^	−0.18 ± 0.01 ^b^	−0.13 ± 0.05 ^a^	**5.42**
		**10%**	−0.20 ± 0.02 ^c^	−0.18 ± 0.02 ^bc^	−0.15 ± 0.02 ^ab^	−0.14 ± 0.01 ^a^	**7.00**
**Date**	** *Firmness* ** ** *(N)* **	**2.5%**	7.95 ± 0.20 ^a^	7.04 ± 0.35 ^b^	6.75 ± 0.15 ^c^	5.96 ± 0.11 ^d^	**69.04**
		**5%**	7.85 ± 0.30 ^a^	6.98 ± 0.06 ^b^	6.45 ± 0.05 ^c^	5.76 ± 0.10 ^d^	**99.91**
		**7.5**%	7.55 ± 0.20 ^a^	6.45 ± 0.15 ^b^	5.85 ± 0.11 ^c^	5.15 ± 0.05 ^d^	**142.61**
		**10%**	7.24 ± 0.18 ^a^	6.19 ± 0.20 ^b^	5.64 ± 0.12 ^c^	5.01 ± 0.10 ^d^	**103.50**
	** *Adhesiveness (N.s)* **	**2.5%**	−0.20 ± 0.03 ^b^	−0.17 ± 0.01 ^ab^	−0.15 ± 0.02 ^a^	−0.13 ± 0.02 ^a^	**5.94**
		**5%**	−0.18 ± 0.01 ^b^	−0.17 ± 0.01 ^b^	−0.16 ± 0.02 ^ab^	−0.14 ± 0.01 ^a^	**5.00**
		**7.5%**	−0.17 ± 0.01 ^c^	−0.15 ± 0.01 ^bc^	−0.14 ± 0.01 ^ab^	−0.12 ± 0.02 ^a^	**7.42**
		**10%**	−0.17 ± 0.01 ^b^	−0.14 ± 0.02 ^a^	−0.12 ± 0.01 ^a^	−0.11 ± 0.02 ^a^	**8.40**
**Control**	** *Firmness(N)* **	**0%**	12.50 ± 0.05 ^a^	10.25 ± 0.15 ^b^	9.55 ± 0.10 ^c^	9.02 ± 0.01 ^d^	**802.25**
	** *Adhesiveness (N.s)* **		−0.96 ± 0.01 ^c^	−0.85 ± 0.03 ^b^	−0.80 ± 0.03 ^a^	−0.76 ± 0.01 ^a^	**28.37**

T_0_: 0 days; T_1_: 7 days; T_2_: 21 days; and T_3_: 30 days of storage. Mean (*n* = 3) ± SD. Means in the same line with different letters are significantly different (*p* < 0.05). DM: dry matter.

**Table 4 foods-11-01393-t004:** Color parameters of enriched pasta during storage.

		Pear				Apple				Date			
		T_0_	T_1_	T_2_	T_3_	T_0_	T_1_	T_2_	T_3_	T_0_	T_1_	T_2_	T_3_
**L***	2.5%	56.24 ±1.72	54.20 ±0.13	54.01 ±0.41	53.01 ±2.01	53.17 ±0.15	53.69 ±0.31	52.25 ±0.21	52.01 ±0.11	53.04 ±0.38	53.65 ±0.56	52.72 ±0.14	52.01 ±0.04
	5%	55.34 ±0.51	53.30 ±1.41	49.44 ±1.20	49.01 ±0.27	53.22 ±1.50	53.70 ±2.01	53.60 ±1.03	52.20 ±1.23	45.79 ±0.46	43.32 ±0.57	44.72 ±0.65	41.25 ±1.75
	7.5%	49.48 ±1.01	47.78 ±2.12	47.21 ±0.01	46.00 ±1.25	52.67 ±0.06	52.55 ±0.75	51.44 ±0.38	50.62 ±0.34	41.61 ±0.08	40.26 ±0.04	40.76 ±0.31	38.54 ±0.02
	10%	41.38 ±1.51	40.23 ±1.26	40.33 ±2.01	38.56 ±1.14	51.86 ±0.10	50.73 ±0.15	50.01 ±0.01	48.41 ±2.02	39.29 ±0.45	38.84 ±0.08	37.20 ±0.75	36.75 ±0.13
**a***	2.5%	4.87 ±0.18	4.60 ±0.06	4.20 ±0.05	4.16 ±0.02	4.27 ±0.45	4.15 ±0.23	4.11 ±0.06	4.12 ±0.02	3.56 ±0.10	3.16 ±0.05	3.17 ±0.01	3.20 ±0.38
	5%	1.53 ±0.03	1.50 ±0.01	1.49 ±0.04	1.42 ±0.02	1.64 ±0.02	1.31 ±0.18	1.11 ±0.07	1.10 ±0.13	1.89 ±0.25	1.56 ±0.36	1.16 ±0.31	1.11 ±0.01
	7.5%	0.46 ±0.02	0.45 ±0.04	0.43 ±0.07	0.44 ±0.01	0.41 ±0.03	0.40 ±0.01	0.40 ±0.04	0.38 ±0.01	0.75 ±0.05	0.71 ±0.02	0.72 ±0.06	0.70 ±0.05
	10%	−1.40 ±0.67	−1.38 ±0.29	−1.39 ±0.38	−1.37 ±0.13	−1.72 ±0.14	−1.70 ±0.04	−1.69 ±0.01	−1.67 ±0.06	−1.56 ±0.02	−1.57 ±0.01	−1.55 ±0.06	−1.52 ±0.08
**b***	2.5%	16.56 ±0.05	16.33 ±0.18	16.28 ±0.03	16.13 ±0.02	18.40 ±0.10	18.14 ±0.01	18.10 ±0.02	18.16 ±0.02	12.78 ±0.05	12.24 ±0.31	12.15 ±0.14	12.10 ±0.01
	5%	13.57 ±0.36	13.36 ±0.76	13.20 ±0.29	13.00 ±0.12	16.32 ±0.07	16.04 ±0.01	15.97 ±0.05	16.00 ±0.02	10.83 ±0.04	10.17 ±0.03	10.11 ±0.20	10.02 ±0.13
	7.5%	10.45 ±0.46	10.25 ±0.11	10.23 ±0.03	10.13 ±0.02	14.70 ±0.03	14.15 ±0.11	14.05 ±0.06	14.00 ±0.02	8.36 ±0.01	8.08 ±0.04	7.97 ±0.01	8.00 ±0.06
	10%	8.59 ±0.22	8.23 ±0.01	8.20 ±0.14	8.25 ±0.24	12.83 ±0.05	12.26 ±0.05	12.13 ±0.01	12.10 ±0.02	5.19 ±0.01	5.10 ±0.31	5.06 ±0.14	5.08 ±0.05
∆E	2.5%	14.14 ±0.03	15.88 ±0.01	16.05 ±0.25	16.96 ±0.15	10.69 ±0.10	15.34 ±0.25	16.63 ±0.21	16.82 ±0.14	22.13 ±0.25	18.95 ±0.10	24.41 ±0.05	20.24 ±0.01
	5%	16.96 ±0.20	18.58 ±0.05	21.70 ±0.10	22.17 ±0.52	13.72 ±0.21	16.65 ±0.18	16.80 ±0.34	17.93 ±0.19	25.99 ±0.23	28.43 ±0.14	27.34 ±0.27	30.30 ±0.05
	7.5%	23.46 ±0.15	24.90 ±0.10	26.96 ±0.23	27.18 ±0.14	16.16 ±0.14	18.83 ±0.28	19.76 ±0.26	20.45 ±0.31	30.92 ±0.25	32.20 ±0.05	31.84 ±0.20	33.69 ±0.34
	10%	31.28 ±0.18	32.42 ±0.30	32.35 ±0.05	33.80 ±0.10	20.14 ±0.15	21.83 ±0.27	22.45 ±0.04	23.71 ±0.16	34.12 ±0.14	35.32 ±0.34	36.67 ±0.25	37.02 ±0.13

T_0_: 0 days; T_1_: 7 days; T_2_: 21 days; and T_3_: 30 days of storage. Mean (*n* = 3) ± SD.

**Table 5 foods-11-01393-t005:** Water activity of by-product-enriched pasta formulations during storage.

By-Product Addition (g/100 g DM)		T_0_	T_1_	T_2_	T_3_	*F*
Apple	2.5%	0.688 ± 0.005 ^a^	0.585 ± 0.002 ^b^	0.573 ± 0.005 ^c^	0.565 ± 0.004 ^d^	**602.20**
	5%	0.686 ± 0.001 ^a^	0.592 ± 0.005 ^b^	0.577 ± 0.001 ^c^	0.568 ± 0.003 ^d^	**986.75**
	7.5%	0.685 ± 0.004 ^a^	0.597 ± 0.003 ^b^	0.580 ± 0.003 ^c^	0.570 ± 0.005 ^d^	**561.22**
	10%	0.701 ± 0.003 ^a^	0.599 ± 0.002 ^b^	0.582 ± 0.005 ^c^	0.571 ± 0.005 ^d^	**677.12**
Pear	2.5%	0.695 ± 0.005 ^a^	0.577 ± 0.006 ^b^	0.568 ± 0.005 ^b^	0.546 ± 0.003 ^c^	**528.18**
	5%	0.699 ± 0.002 ^a^	0.579 ± 0.003 ^b^	0.569 ± 0.002 ^c^	0.549 ± 0.001 ^d^	**3066.66**
	7.5%	0.701 ± 0.005 ^a^	0.581 ± 0.002 ^b^	0.572 ± 0.005 ^bc^	0.550 ± 0.004 ^c^	**9.80**
	10%	0.702 ± 0.002 ^a^	0.583 ± 0.001 ^b^	0.574 ± 0.005 ^c^	0.555 ± 0.001 ^d^	**1721.93**
Date	2.5%	0.689 ± 0.003 ^a^	0.568 ± 0.001 ^b^	0.545 ± 0.005 ^c^	0.540 ± 0.004 ^c^	**115.21**
	5%	0.691 ± 0.005 ^a^	0.690 ± 0.001 ^a^	0.573 ± 0.004 ^b^	0.544 ± 0.002 ^c^	**1551.73**
	7.5%	0.698 ± 0.005 ^a^	0.590 ± 0.006 ^b^	0.576 ± 0.002 ^b^	0.551 ± 0.001 ^c^	**204.59**
	10%	0.700 ± 0.003 ^a^	0.598 ± 0.005 ^b^	0.579 ± 0.001 ^b^	0.555 ± 0.002 ^c^	**239.85**
Control	0%	0.672 ± 0.001 ^a^	0.558 ± 0.001 ^b^	0.541 ± 0.005 ^b^	0.537 ± 0.003 ^b^	**16.24**

T_0_: 0 days; T_1_: 7 days; T_2_: 21 days; and T_3_: 30 days of storage. Mean (*n* = 3) ± SD. Means in the same line with different letters are significantly different (*p* < 0.05).

**Table 6 foods-11-01393-t006:** Alveograph characteristics of dough containing different rates and kinds of fiber by-products.

		Tenacity (P) (mm of H_2_O)	Extensibility (L) (mm)	P/L	Deformation Energy (×10^−4^)
** *Dough Control* **	0%	80.23 ± 1.80 ^h^	62.53 ± 0.40 ^a^	1.29 ± 0.05 ^g^	154.82 ± 2.30 ^a^
** *Flour with Date Fibers* **	2.5%	115.23 ± 1.50 ^g^	25.84 ± 0.50 ^b^	4.46 ± 0.21 ^f^	135.21 ± 1.15 ^b^
	5%	123.16 ± 1.50 ^ef^	23.64 ± 1.20 ^bcd^	5.21 ± 0.03 ^e^	133.11 ± 2.23 ^bc^
	7.5%	130.52 ± 2.30 ^bc^	20.15 ± 1.40 ^defg^	6.47 ± 0.01 ^c^	130.50 ± 1.01 ^cde^
	10%	135.43 ± 1.20 ^a^	18.01 ± 1.50 ^fg^	7.52 ± 0.22 ^a^	128.30 ± 2.84 ^ef^
** *Flour with Pear Fibers* **	2.5%	114.12 ± 1.40 ^g^	26.23 ± 1.50 ^b^	4.35 ± 0.10 ^f^	133.10 ± 1.5 0 ^bcd^
	5%	122.40 ± 2.80 ^ef^	23.60 ± 3.70 ^bcd^	5.18 ± 0.13 ^e^	129.50 ± 5.30 ^def^
	7.5%	128.00 ± 1.50 ^cd^	21.54 ± 1.10 ^cdef^	5.94 ± 0.23 ^d^	127.20 ± 2.10 ^efg^
	10%	133.60 ± 3.50 ^ab^	18.76 ± 3.80 ^fg^	7.12 ± 0.01 ^b^	125.23 ± 1.40 ^fg^
** *Flour with Apple Fibers* **	2.5%	121.23 ± 3.50 ^f^	24.13 ± 1.40 ^bc^	5.02 ± 0.11 ^e^	134.23 ± 2.80 ^bc^
	5%	126.30 ± 2.10 ^de^	22.10 ± 1.70 ^cde^	5.71 ± 0.02 ^d^	130.50 ± 3.50 ^bc^
	7.5%	130.23 ± 1.80 ^bc^	19.50 ± 1.20 ^efg^	6.68 ± 0.22 ^c^	126.23 ± 2.40 ^efg^
	10%	135.10 ± 2.70 ^a^	17.50 ± 1.90 ^g^	7.72 ± 0.01 ^a^	124.10 ± 1.50 ^g^
** *F* **		**142.84**	**118.49**	**531.40**	**34.47**

Means in the same column with different letters are significantly different (*p* < 0.05).

**Table 7 foods-11-01393-t007:** Thermal gelling properties of pasta dough.

	By-Product Addition (g/100 g DM)	T _onset_ (°C)	T _endset_ (°C)	Enthalpy (∆H j/g)	Gelatinization Temperature (°C)
**Control**	0%	57.23 ± 0.25 ^a^	67.42 ± 0.29 ^c^	0.97 ± 0.06 ^a^	62.07 ± 0.22 ^bc^
**Pear**	2.5%	56.03 ± 0.40 ^b^	67.89 ± 0.25 ^bc^	0.80 ± 0.01 ^bcd^	62.31 ± 0.16 ^c^
	5%	55.56 ± 0.45 ^bc^	68.06 ± 0.35 ^bc^	0.71 ± 0.03 ^ef^	62.54 ± 0.05 ^bc^
	7.5%	55.30 ± 0.30 ^cd^	68.35 ± 0.15 ^abc^	0.66 ± 0.03 ^fg^	62.61 ± 0.05 ^abc^
	10%	55.05 ± 0.21 ^cd^	68.20 ± 0.65 ^bc^	0.62 ± 0.01 ^g^	62.84 ± 0.20 ^a^
**Apple**	2.5%	56.88 ± 0.43 ^a^	68.01 ± 0.12 ^bc^	0.84 ± 0.02 ^b^	61.10 ± 0.35 ^d^
	5%	56.01 ± 0.31 ^b^	68.25 ± 0.35 ^abc^	0.76 ± 0.01 ^cde^	62.29 ± 0.12 ^c^
	7.5%	55.50 ± 0.26 ^bc^	68.45 ± 0.10 ^c^	0.68 ± 0.02 ^de^	62.36 ± 0.23 ^c^
	10%	55.25 ± 0.29 ^cd^	68.81 ± 0.20 ^a^	0.65 ± 0.03 ^fg^	62.44 ± 0.25 ^bc^
**Date**	2.5%	55.86 ± 0.32 ^b^	67.95 ± 0.14 ^bc^	0.82 ± 0.01 ^bc^	62.37 ± 0.30 ^c^
	5%	55.10 ± 0.15 ^cd^	68.08 ± 0.23 ^bc^	0.74 ± 0.02 ^cde^	62.56 ± 0.15 ^abc^
	7.5%	54.89 ± 0.10 ^d^	68.55 ± 0.15 ^ab^	0.64 ± 0.02 ^g^	62.78 ± 0.11 ^abc^
	10%	54.75 ± 0.19 ^d^	68.70 ± 0.22 ^ab^	0.60 ± 0.01 ^g^	62.93 ± 0.20 ^ab^
** *F* **		**19.10**	**2.75**	**21.95**	**9.08**

Means in the same column with different letters are significantly different (*p* < 0.05).

**Table 8 foods-11-01393-t008:** Sensory attributes of cooked pasta.

		Taste	Aftertaste	Appearance	Texture	Color	Overall Acceptability
**Control**	**0%**	6.5 ± 0.4 ^a^	6.6 ± 0.5 ^c^	6.0 ± 0.3 ^bcd^	6.3 ± 0.2 ^ab^	5.9 ± 0.4 ^bc^	6.2 ± 0.2 ^abc^
**Flour with Apple Fibers**	**2.5%**	6.6 ± 0.1 ^a^	6.5 ± 0.2 ^ab^	6.2 ± 0.1 ^ab^	6.5 ± 0.3 ^a^	6.4 ± 0.1 ^a^	6.4 ± 0.1 ^a^
	**5%**	6.5 ± 0.3 ^a^	6.0 ± 0.1 ^c^	6.0 ± 0.2 ^bcd^	6.0 ± 0.1 ^bcd^	5.9 ± 0.1 ^b^	6.0 ± 0.1 ^cde^
	**7.5%**	6.3 ± 0.2 ^ab^	6.1 ± 0.2 ^bc^	5.9 ± 0.3 ^be^	5.4 ± 0.4 ^fg^	5.6 ± 0.2 ^cde^	5.8 ± 0.2 ^ef^
	**10%**	6.4 ± 0.1 ^ab^	6.0 ± 0.1 ^c^	5.8 ± 0.1 ^cde^	5.2 ± 0.2 ^g^	5.4 ± 0.2 ^e^	5.7 ± 0.1 ^f^
**Flour with Pear Fibers**	**2.5%**	6.4 ± 0.1 ^ab^	6.5 ± 0.2 ^ab^	6.1 ± 0.1 ^abc^	6.2 ± 0.2 ^abc^	6.5 ± 0.1 ^a^	6.3 ± 0.1 ^ab^
	**5%**	6.2 ± 0.1 ^abc^	6.0 ± 0.2 ^c^	5.8 ± 0.1 ^cde^	5.9 ± 0.1 ^cde^	6.0 ± 0.1 ^bc^	5.9 ± 0.2 ^def^
	**7.5%**	6.0 ± 0.2 ^bc^	6.1 ± 0.1 ^bc^	5.6 ± 0.2 ^e^	5.6 ± 0.1 ^ef^	5.8 ± 0.1 ^bcd^	5.8 ± 0.2 ^ef^
	**10%**	5.8 ± 0.3 ^c^	6.0 ± 0.2 ^c^	5.7 ± 0.1 ^de^	5.5 ± 0.1 ^fg^	5.5 ± 0.1 ^de^	5.8 ± 0.1 ^ef^
**Flour with Date Fibers**	**2.5%**	6.3 ± 0.1 ^ab^	6.5 ± 0.1 ^ab^	6.4 ± 0.1 ^a^	6.4 ± 0.2 ^a^	6.5 ± 0.1 ^a^	6.4 ± 0.1 ^a^
	**5%**	6.2 ± 0.2 ^abc^	6.4 ± 0.2 ^abc^	6.2 ± 0.1 ^ab^	5.7 ± 0.2 ^def^	6.0 ± 0.2 ^b^	6.1 ± 0.1 ^bcd^
	**7.5%**	6.4 ± 0.1 ^ab^	6.4 ± 0.1 ^abc^	6.0 ± 0.2 ^bcd^	5.6 ± 0.1 ^ef^	5.7 ± 0.1 ^bcde^	6.2 ± 0.1 ^abc^
	**10%**	6.3 ± 0.1 ^abc^	6.0 ± 0.3 ^c^	5.8 ± 0.2 ^cde^	5.4 ± 0.1 ^fg^	5.5 ± 0.2 ^de^	5.8 ± 0.2 ^ef^
** *F* **		**2.60**	**3.66**	**4.84**	**14.05**	**13.81**	**8.67**

Means in the same column with different letters are significantly different (*p* < 0.05).

## Data Availability

Not applicable.

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
