# Peer review of "Date, Apple, and Pear By-Products as Functional Ingredients in Pasta: Cooking Quality Attributes and Physicochemical, Rheological, and Sensorial Properties"

_foods, 2022, doi:10.3390/foods11101393_

Round 1

Reviewer 1 Report

Totally, the research has been well designed but the writing and presentation of the results are not good. Furthermore, the English is not good enough and needs to revise.

The paper lacks novelty.

The introduction has been written poor.

The title is too long, should be shorter.

There are many error and mistake (space between words) present in abstract. Extensive editing of English language and style required for all the text.

Line 29: greenness is -a*, not a*

Line 28, 29: where is the result of this sentence “Cooked pasta samples resulted in significantly (p<0.05) darker (L*) and more greenness (a*) than control pasta”? What does it means?

Reviewer 2 Report

The work concerns various fruit fibers and their influence on selected functional properties of pasta, so I believe that the keywords are completely wrong. they do not relate to the subject of the work, but mention analytical methods. Keywords should be corrected. 

The first part of the "introduction" and its content are relevant to the title of the work. The part about waste management and its global production takes too long. should be shortened. The aim of the work should be clearly specified, not the methods and determinations that have been made. 

  • 3. Results and discussion - text formatting needed to be improved, too large line spacing 
  • table 1,2,5,6,7,8 - letters denoting Means in column with different letters are significantly different (P <0.05), should be after the value of the standard deviation and not the result
  • subsections analyzing physical properties 2.4. - all sub-items 2.4.1.1-4 - the description of calculations should be separated either as equal to the verbal description or as an equation with abbreviations. This will significantly facilitate the reader's perception. 
  • a reference to table 1 in the text should be before the table, not the other way around 

Reviewer 3 Report

This article investigates the impact of incorporating pear, date, and apple by-products addition on pasta properties. The experimental campaign was planned and carried out really well, without neglecting any aspect of the characterization of the dough and the final product. The results are interesting and the conclusions are supported by them. Additionally, the article is well written. There are just few suggestions that I recommended in the attached file and that can further improve the paper. Also have a look at the format and style of the paper that in some points does not follow the journal template.

Reviewer 4 Report

Please make sure that all equipment or software includes the country of origin and manufacturer - e.g. line 210 is missing that information

Section 3.1

%<% is extremely confusing here, what is the author trying to imply?

ANOVAs - It doesn't make sense if the authors merged all the comparisons in one column, this needs to be separated where the effect is compared within each fiber pulp type. Then all results will be significantly different from control. Also incude F values for each analyses.

Table 3-5 now includes another factor, time. Did the author attempt to run 2way ANOVA on this, or what is the aim?

There is A LOT of results that the authors collected which is great. But it is highly recommended to justify why these results are collected and how it impacts sensory and consumer acceptance.

Figure 2. Although running a PCA might be tempting, there are multiple data tables here and therefore a decomposition is required prior to 'merging' them together. I'd recommend a MFA instead.

Discussion is rather cluttered and embedded in the results section - while this is fine, I'd recommend the authors to extract the key highlight of their results and discuss accordingly.

Also include limitations and future research area for this.

Round 2

Reviewer 1 Report

I strongly suggest you to use a native enslish editor! Many mistakes related to language and English are remained! for example: line 293, 255, 259, 318, 319, 403, ......

Please check all the statistical anlysis letters in tables, some of them are wrong!

P>0.05 should be replace to p>0.05, please apply in all the text.

The inside borders of tables (horizontal borders) must be eliminate.

Reviewer 4 Report

I agree that the authors are trying to compare their results with the control in mind. But by doing so the authors are merging their factors and had made it somewhat difficult to see how different by-product addition affects their measures. 

Also include the F values for the analysis performed just to have an indication the significance of effect of these tables.

If MFA produces similar results to PCA, then please use MFA as it is more statistically correct.
